# Quantum Perceptron Models

**Nathan Wiebe**
Microsoft Research
Redmond WA, 98052
nawiebe@microsoft.com

**Ashish Kapoor**
Microsoft Research
Redmond WA, 98052
akapoor@microsoft.com

**Krysta M Svore**
Microsoft Research
Redmond WA, 98052
ksvore@microsoft.com

## Abstract

We demonstrate how quantum computation can provide non-trivial improvements in the computational and statistical complexity of the perceptron model. We develop two quantum algorithms for perceptron learning. The first algorithm exploits quantum information processing to determine a separating hyperplane using a number of steps sublinear in the number of data points $N$, namely $O(\sqrt{N})$. The second algorithm illustrates how the classical mistake bound of $O(\frac{1}{\gamma^2})$ can be further improved to $O(\frac{1}{\sqrt{\gamma}})$ through quantum means, where $\gamma$ denotes the margin. Such improvements are achieved through the application of quantum amplitude amplification to the version space interpretation of the perceptron model.

## 1 Introduction

Quantum computation is an emerging technology that utilizes quantum effects to achieve significant, and in some cases exponential, speed-ups of algorithms over their classical counterparts. The growing importance of machine learning has in recent years led to a host of studies that investigate the promise of quantum computers for machine learning [1, 2, 3, 4, 5, 6, 7, 8, 9].

While a number of important quantum speedups have been found, the majority of these speedups are due to replacing a classical subroutine with an equivalent albeit faster quantum algorithm. The true potential of quantum algorithms may therefore remain underexploited since quantum algorithms have been constrainted to follow the same methodology behind traditional machine learning methods [10, 8, 9]. Here we consider an alternate approach: we devise a new machine learning algorithm that is tailored to the speedups that quantum computers can provide.

We illustrate our approach by focusing on perceptron training [11]. The perceptron is a fundamental building block for various machine learning models including neural networks and support vector machines [12]. Unlike many other machine learning algorithms, tight bounds are known for the computational and statistical complexity of traditional perceptron training. Consequently, we are able to rigorously show different performance improvements that stem from either using quantum computers to improve traditional perceptron training or from devising a new form of perceptron training that aligns with the capabilities of quantum computers.

We provide two quantum approaches to perceptron training. The first approach focuses on the computational aspect of the problem and the proposed method quadratically reduces the scaling of the complexity of training with respect to the number of training vectors. The second algorithm focuses on statistical efficiency. In particular, we use the mistake bounds for traditional perceptron training methods and ask if quantum computation lends any advantages. To this end, we propose an algorithm that quadratically improves the scaling of the training algorithm with respect to the margin between the classes in the training data. The latter algorithm combines quantum amplitude estimation in the version space interpretation of the perceptron learning problem. Our approaches showcase the trade-offs that one can consider in developing quantum algorithms, and the ultimate advantages of performing learning tasks on a quantum computer.

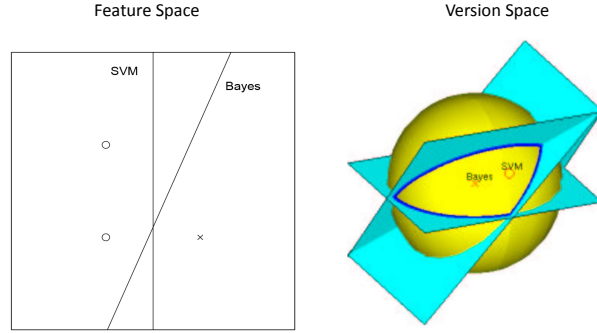

Figure 1: Version space and feature space views of classification. This figure is from [18].

The rest of the paper is organized as follows: we first cover the background on perceptrons, version space and Grover search. We then present our two quantum algorithms and provide analysis of their computational and statistical efficiency before concluding.

## 2 Background

### 2.1 Perceptrons and Version Space

Given a set of $N$ separable training examples $\{\phi_1, .., \phi_N\} \in \mathbb{R}^D$ with corresponding labels $\{y_1, .., y_N\}$, $y_i \in \{+1, -1\}$, the goal of perceptron learning is to recover a hyperplane $w$ that perfectly classifies the training set [11]. Formally, we want $w$ such that $y_i \cdot w^T \phi_i > 0$ for all $i$. There are various simple online algorithms that start with a random initialization of the hyperplane and make updates as they encounter more and more data [11, 13, 14, 15]; however, the rule that we consider for online perceptron training is, upon misclassifying a vector $(\phi, y)$, $w \leftarrow w + y\phi$.

A remarkable feature of the perceptron model is that upper bounds exist for the number of updates that need to be made during this training procedure. In particular, if the training data is composed of unit vectors, $\phi_i \in \mathbb{R}^D$, that are separated by a margin of $\gamma$ then there are perceptron training algorithms that make at most $O(\frac{1}{\gamma^2})$ mistakes [16], independent of the dimension of the training vectors. Similar bounds also exist when the data is not separated [17] and also for other generalizations of perceptron training [13, 14, 15]. Note that in the worst case, the algorithm will need to look at all points in the training set at least once, consequently the computation complexity will be $O(N)$.

Our goal is to explore if the quantum procedures can provide improvements both in terms of computational complexity (that is better than $O(N)$) and statistical efficiency (improve upon $O(\frac{1}{\gamma^2})$). Instead of solely applying quantum constructs to the feature space, we also consider the version space interpretation of perceptrons which leads to the improved scaling with $\gamma$.

Formally, version space is defined as the set of all possible hyperplanes that perfectly separate the data: $VS := \{w | y_i \cdot w^T \phi_i > 0 \text{ for all } i\}$. Given a training datum, the traditional representation is to depict data as points in the feature space and use hyperplanes to depict the classifiers. However, there exists a dual representation where the hyperplanes are depicted as points and the data points are represented as hyperplanes that induce constraints on the feasible set of classifiers. Figure 1, which is borrowed from [18], illustrates the version space interpretation of perceptrons. Given three labeled data points in a 2D space, the dual space illustrates the set of normalized hyperplanes as a yellow ball with unit radius. The third dimension corresponds to the weights that multiply the two dimensions of the input data and the bias term. The planes represent the constraints imposed by observing the labeled data as every labeled data renders one-half of the space infeasible. The version space is then the intersection of all the half-spaces that are valid. Naturally, classifiers including SVMs [12] and Bayes point machines [19] lie in the version space.

We note that there are quantum constructs such as Grover search and amplitude amplification which provide non-trivial speedups for the search task. This is the main reason why we resort to the version space interpretation. We can use this formalism to simply pose the problem of determining the

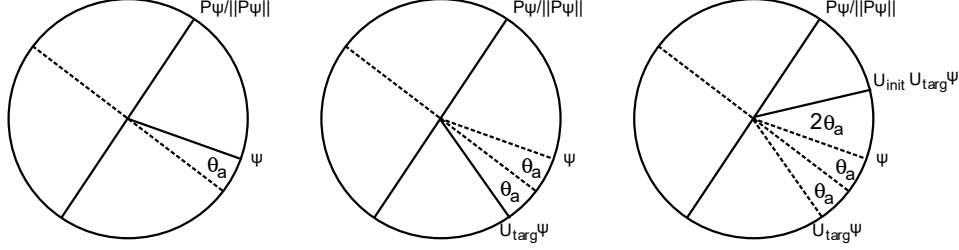

Figure 2: A geometric description of the action of $U_{\text{grover}}$ on an initial state vector $\psi$.

separating hyperplane as a search problem in the dual space. For example given a set of candidates hyperplanes, our problem reduces to searching amongst the sample set for the classifier that will successfully classify the entire set. Therefore training the perceptron is equivalent to finding *any* feasible point in the version space. We describe these quantum constructs in detail below.

## 2.2 Grover's Search

Both quantum approaches introduced in this work and their corresponding speed-ups stem from a quantum subroutine called Grover's search [20, 21], which is a special case of a more general method referred to as amplitude amplification [22]. Rather than sampling from a probability distribution until a given marked element is found, the Grover search algorithm draws only one sample and then uses quantum operations to modify the distribution from which it sampled. The probability distribution is rotated, or more accurately the quantum state that yields the distribution is rotated, into one whose probability is sharply concentrated on the marked element. Once a sharply peaked distribution is identified, the marked item can be found using just one sample. In general, if the probability of finding such an element is known to be $a$ then amplitude amplification requires $O(\sqrt{1/a})$ operations to find the marked item with certainty.

While Grover's search is a *quantum subroutine*, it can in fact be understood using only geometric arguments. The only notions from quantum mechanics used are those of the quantum state vector and that of Born's rule (measurement). A quantum state vector is a complex unit vector whose components have magnitudes that are equal to the square–roots of the probabilities. In particular, if $\psi$ is a quantum state vector and $p$ is the corresponding probability distribution then

$$p = \psi^\dagger \circ \psi, \tag{1}$$

where the unit column vector $\psi$ is called the quantum state vector which sits in the vector space $\mathbb{C}^n$, $\circ$ is the Hadamard (pointwise) product and $\dagger$ is the complex conjugate transpose. A quantum state can be measured such that if we have a quantum state vector $\psi$ and a basis vector $w$ then the probability of measuring $\psi = w$ is $|\langle \psi, w \rangle|^2$, where $\langle \cdot, \cdot \rangle$ denotes the inner product.

We need to implement two unitary operations in order to perform the search algorithm:

$$U_{\text{init}} = 2\psi\psi^\dagger - \mathbb{1}, U_{\text{targ}} = \mathbb{1} - 2P. \tag{2}$$

The operators $U_{\text{init}}$ and $U_{\text{targ}}$ can be interpreted geometrically as reflections within a two–dimensional space spanned by the vectors $\psi$ and $P\psi$. If we assume that $P\psi \neq 0$ and $P\psi \neq \psi$ then these two reflection operations can be used to rotate $\psi$ in the space $\text{span}(\psi, P\psi)$. Specifically this rotation is $U_{\text{grover}} = U_{\text{init}} U_{\text{targ}}$. Its action is illustrated in Figure 2. If the angle between the vector $\psi$ and $P\psi/\|P\psi\|$ is $\pi/2 - \theta_a$, where $\theta_a := \sin^{-1}(|\langle \psi, P\psi/\|P\psi\| \rangle|)$. It then follows from elementary geometry and the rule for computing the probability distribution from a quantum state (known as Born's rule) that after $j$ iterations of Grover's algorithm the probability of finding a desirable outcome is

$$p(\psi \in \nu_{\text{good}} | j) = \sin^2((2j+1)\theta_a). \tag{3}$$

It is then easy to see that if $\theta_a \ll 1$ and a probability of success greater than $1/4$ is desired then $j \in O(1/\sqrt{\theta_a})$ suffices to find a marked outcome. This is quadratically faster than is possible from statistical sampling, which requires $O(1/\theta_a)$ samples on average. Simple modifications to this algorithm allow it to be used in cases where $\theta_a$ is not known [21, 22].

# 3   Online quantum perceptron

Now that we have discussed Grover's search we turn our attention to applying it to speed up online perceptron training. In order to do so, we first need to define the quantum model that we wish to use as our quantum analogue of perceptron training. While there are many ways of defining such a model but the following approach is perhaps the most direct. Although the traditional feature space perceptron training algorithm is online [16], meaning that the training examples are provided one at a time to it in a streaming fashion, we deviate from this model slightly by instead requiring that the algorithm be fed training examples that are, in effect, sampled uniformly from the training set. This is a slightly weaker model, as it allows for the possibility that some training examples will be drawn multiple times. However, the ability to draw quantum states that are in a uniform superposition over all vectors in the training set enables quantum computing to provide advantages over both classical methods that use either access model.

We assume without loss of generality that the training set consists of $N$ unit vectors, $\phi_1, \ldots, \phi_N$. If we then define $\Phi_1, \ldots, \Phi_N$ to be the basis vectors whose indices each coincide with a $(B+1)$-bit representation of the corresponding $(\phi_j, y_j)$ where $y_j \in \{-1, 1\}$ is the class assigned to $\phi_j$ and let $\Phi_0$ be a fixed unit vector that is chosen to represent a blank memory register.

We introduce the vectors $\Phi_j$ to make it clear that the quantum vectors states used to represent training vectors do not live in the same vector space as the training vectors themselves. We choose the quantum state vectors here to occupy a larger space than the training vectors because the Heisenberg uncertainty principle makes it much more difficult for a quantum computer to compute the class that the perceptron assigns to a training vector in such cases.

For example, the training vector $(\phi_j, y_j) \equiv ([0, 0, 1, 0]^T, 1)$ can be encoded as an unsigned integer $00101 \equiv 5$, which in turn can be represented by the unit vector $\Phi = [0, 0, 0, 0, 0, 1]^T$. More generally, if $\phi_j \in \mathbb{R}^D$ were a vector of floating point numbers then a similar vector could be constructed by concatenating the binary representations of the $D$ floating point numbers that comprise it with $(y_j + 1)/2$ and express the bit string as an unsigned integer, $Q$. The integer can then be expressed as a unit vector $\Phi : [\Phi]_q = \delta_{q,Q}$. While encoding the training data as an exponentially long vector is inefficient in a classical computer, it is not in a quantum computer because of the quantum computer's innate ability to store and manipulate exponentially large quantum state vectors.

Any machine learning algorithm, be it quantum or classical, needs to have a mechanism to access the training data. We assume that the data is accessed via an oracle that not only accesses the training data but also determines whether the data is misclassified. To clarify, let $\{u_j : j = 1 : N\}$ be an orthonormal basis of quantum state vectors that serve as addresses for the training vectors in the database. Given an input address for the training datum, the unitary operations $U$ and $U^\dagger$ allow the quantum computer to access the corresponding vector. Specifically, for all $j$

$$U[u_j \otimes \Phi_0] \quad = u_j \otimes \Phi_j, \qquad U^\dagger[u_j \otimes \Phi_j] \quad = u_j \otimes \Phi_0. \tag{4}$$

Given an input address vector $u_j$, the former corresponds to a database access and the latter inverts the database access.

Note that because $U$ and $U^\dagger$ are linear operators we have that $U \sum_{j=1}^{N} u_j \otimes \Phi_0 = \sum_j u_j \otimes \Phi_j$. A quantum computer can therefore access each training vector simultaneously using a single operation. The resultant vector is often called in the physics literature a *quantum superposition of states* and this feature of linear transformations is referred to as quantum parallelism within quantum computing.

The next ingredient that we need is a method to test if the perceptron correctly assigns a training vector addressed by a particular $u_j$. This process can be pictured as being performed by a unitary transformation that flips the sign of any basis-vector that is misclassified. By linearity, a single application of this process flips the sign of any component of the quantum state vector that coincides with a misclassified training vector. It therefore is no more expensive than testing if a given training vector is misclassified in a classical setting. We denote the operator, which depends on the perceptron weights $w$, $F_w$ and require that

$$F_w[u_j \otimes \Phi_0] = (-1)^{f_w(\phi_j, y_j)}[u_j \otimes \Phi_0], \tag{5}$$

where $f_w(\phi_j)$ is a Boolean function that is 1 if and only if the perceptron with weights $w$ misclassifies training vector $\phi_j$. Since the classification step involves computing the dot–products of finite size vectors, this process is efficient given that the $\Phi_j$ are efficiently computable.

We apply $F_w$ in the following way. Let $\mathcal{F}_w$ be a unitary operation such that

$$\mathcal{F}_w \Phi_j = (-1)^{f_w(\phi_j, y_j)} \Phi_j. \tag{6}$$

$\mathcal{F}_w$ is easy to implement in the quantum computer using a multiply controlled phase gate and a quantum implementation of the perceptron classification algorithm, $f_w$. We can then write

$$F_w = U^\dagger (\mathbb{1} \otimes \mathcal{F}_w) U. \tag{7}$$

Classifying the data based on the phases (the minus signs) output by $F_w$ naturally leads to a very memory efficient training algorithm because only one training vector is ever stored in memory during the implementation of $F_w$ given in Eq. (7). We can then use $F_w$ to perform Grover's search algorithm, by taking $U_{\text{targ}} = F_w$ and $U_{\text{init}} = 2\psi\psi^\dagger - \mathbb{1}$ with $\psi = \Psi := \frac{1}{\sqrt{N}} \sum_{j=1}^N u_j$, to seek out training vectors that the current perceptron model misclassifies. This leads to a quadratic reduction in the number of times that the training vectors need to be accessed by $F_w$ or its classical analogue.

In the classical setting, the natural object to query is slightly different. The oracle that is usually assumed in online algorithms takes the form $U^c : \mathbb{Z} \mapsto \mathbb{C}^D$ where $U^c(j) = \phi_j$. We will assume that a similar function exists in both the classical and the quantum settings for simplicity. In both cases, we will consider the cost of a query to $U^c$ to be proportional to the cost of a query to $F_w$.

We use these operations in to implement a quantum search for training vectors that the perceptron misclassifies. This leads to a quadratic speedup relative to classical methods as shown in the following theorem. It is also worth noting that our algorithm uses a slight variant on the Grover search algorithm to ensure that the runtime is finite.

**Theorem 1.** *Given a training set that consists of unit vectors $\Phi_1, \ldots, \Phi_N$ that are separated by a margin of $\gamma$ in feature space, the number of applications of $F_w$ needed to infer a perceptron model, $w$, such that $P(\exists j : f_w(\phi_j) = 1) \leq \epsilon$ using a quantum computer is $N_{\text{quant}}$ where*

$$\Omega(\sqrt{N}) \ni N_{\text{quant}} \in O\left(\frac{\sqrt{N}}{\gamma^2} \log\left[\frac{1}{\epsilon\gamma^2}\right]\right),$$

*whereas the number of queries to $f_w$ needed in the classical setting, $N_{\text{class}}$, where the training vectors are found by sampling uniformly from the training data is bounded by*

$$\Omega(N) \ni N_{\text{class}} \in O\left(\frac{N}{\gamma^2} \log\left[\frac{1}{\epsilon\gamma^2}\right]\right).$$

We assume in Theorem 1 that the training data in the classical case is accessed in a manner that is analogous to the sampling procedure used in the quantum setting. If instead the training data is supplied by a stream (as in the standard online model) then the upper bound changes to $N_{\text{class}} \in O(N/\gamma^2)$ because all $N$ training vectors can be deterministically checked to see if they are correctly classified by the perceptron. A quantum advantage is therefore obtained if $N \gg \log^2(1/\epsilon\gamma^2)$.

In order to prove Theorem 1 we need to have two technical lemmas (proven in the supplemental material). The first bounds the complexity of the classical analogue to our training method:

**Lemma 1.** *Given only the ability to sample uniformly from the training vectors, the number of queries to $f_w$ needed to find a training vector that the current perceptron model fails to classify correctly, or conclude that no such example exists, with probability $1 - \epsilon\gamma^2$ is at most $O(N \log(1/\epsilon\gamma^2))$.*

The second proves the correctness of our online quantum perceptron algorithm and bounds the complexity of the algorithm:

**Lemma 2.** *Assuming that the training vectors $\{\phi_1, \ldots, \phi_N\}$ are unit vectors and that they are drawn from two classes separated by a margin of $\gamma$ in feature space, Algorithm 2 will either update the perceptron weights, or conclude that the current model provides a separating hyperplane between the two classes, using a number of queries to $F_w$ that is bounded above by $O(\sqrt{N} \log(1/\epsilon\gamma^2))$ with probability of failure at most $\epsilon\gamma^2$.*

After stating these results, we can now provide the proof of Theorem 1.

*Proof of Theorem 1.* The upper bounds follow as direct consequences of Lemma 2 and Lemma 1. Novikoff's theorem [16, 17] states that the algorithms described in both lemmas must be applied at most $1/\gamma^2$ times before finding the result. However, either the classical or the quantum algorithm may fail to find a misclassified vector at each of the $O(1/\gamma^2)$ steps. The union bound states that the probability that this happens is at most the sum of the respective probabilities in each step. These probabilities are constrained to be $\gamma^2\epsilon$, which means that the total probability of failing to correctly find a mistake is at most $\epsilon$ if both algorithms are repeated $1/\gamma^2$ times (which is the worst case number of times that they need to be repeated).

The lower bound on the quantum query complexity follows from contradiction. Assume that there exists an algorithm that can train an arbitrary perceptron using $o(\sqrt{N})$ query operations. Now we want to show that unstructured search with one marked element can be expressed as a perceptron training algorithm. Let $w$ be a known set of perceptron weights and assume that the perceptron only misclassifies one vector $\phi_1$. Thus if perceptron training succeeds then $w$ the value of $\phi_1$ can be extracted from the updated weights. This training problem is therefore equivalent to searching for a misclassified vector. Now let $\phi_j = [1 \oplus F(j), F(j)]^T \otimes \chi_j$ where $\chi_j$ is a unit vector that represents the bit string $j$ and $F(j)$ is a Boolean function. Assume that $F(0) = 1$ and $F(j) = 0$ if $j \neq 0$, which is without loss of generality equivalent to Grover's problem [20, 21]. Now assume that $\phi_j$ is assigned to class $2F(j) - 1$ and take $w = [1/\sqrt{2}, 1/\sqrt{2}]^T \otimes \frac{1}{\sqrt{N}} \sum_j \chi_j$. This perceptron therefore misclassifies $\phi_0$ and no other vector in the training set. Updating the weights yields $\phi_j$, which in turn yields the value of $j$ such that $F(j) = 1$, and so Grover's search reduces to perceptron training.

Since Grover's search reduces to perceptron training in the case of one marked item the lower bound of $\Omega(\sqrt{N})$ queries for Grover's search [21] applies to perceptron training. Since we assumed that perceptron training needs $o(\sqrt{N})$ queries this is a contradiction and the lower bound must be $\Omega(\sqrt{N})$.

We have assumed that in the classical setting that the user only has access to the training vectors through an oracle that is promised to draw a uniform sample from $\{(\phi_1, y_1), \ldots, (\phi_N, y_N)\}$. Since we are counting the number of queries to $f_w$ it is clear that in the worst possible case that the training vector that the perceptron makes a mistake on can be the last unique value sampled from this list. Thus the query complexity is $\Omega(N)$ classically. □

## 4 Quantum version space perceptron

The strategy for our quantum version space training algorithm is to pose the problem of determining a separating hyperplane as search. Specifically, the idea is to first generate $K$ sample hyperplanes $w_1, \ldots, w_K$ from a spherical Gaussian distribution $\mathcal{N}(0, \mathbb{1})$. Given a large enough $K$, we are guaranteed to have at least one hyperplane amongst the samples that would lie in the version space and perfectly separate the data. As discussed earlier Grover's algorithm can provide quadratic speedup over the classical search consequently the efficiency of the algorithm is determined by $K$. Theorem 2 provides an insight on how to determine this number of hyperplanes to be sampled.

**Theorem 2.** *Given a training set that consists of $d$-dimensional unit vectors $\Phi_1, \ldots, \Phi_N$ with labels $y_1, \ldots, y_N$ that are separated by a margin of $\gamma$ in feature space, then a $D$-dimensional vector $w$ sampled from $\mathcal{N}(0, \mathbb{1})$ perfectly separates the data with probability $\Theta(\gamma)$.*

The proof of this theorem is provided in the supplementary material. The consequence of Theorem 2 stated below is that the expected number of samples $K$, required such that a separating hyperplane exists in the set, only needs to scale as $O(\frac{1}{\gamma})$. This is remarkable because, similar to Novikoff's theorem [16], the number of samples needed does not scale with $D$. Thus Theorem 2 implies that if amplitude amplification is used to boost the probability of finding a vector in the version space then the resulting quantum algorithm will need only $O(\frac{1}{\sqrt{\gamma}})$ quantum steps on average.

Next we show how to use Grover's algorithm to search for a hyperplane that lies in the version space. Let us take $K = 2^\ell$, for positive integer $\ell$. Then given $w_1, \ldots, w_K$ the sampled hyperplanes, we represent $W_1, \ldots, W_K$ to be vectors that encode a binary representation of these random perceptron vectors. In analogy to $\Phi_0$, we also define $W_0$ to be a vector that represents an empty data register. We define the unitary operator $V$ to generate these weights given an address vector $u_j$ using the following

$$V[u_j \otimes W_0] = [u_j \otimes W_j]. \tag{8}$$

In this context we can also think of the address vector, $u_j$, as representing a seed for a pseudo–random number generator that yields perceptron weights $W_j$.

Also let us define the classical analogue of $V$ to be $V^c$ which obeys $V^c(j) = w_j$. Now using $V$ (and applying the Hadamard transform [23]) we can prepare the following quantum state

$$\Psi := \frac{1}{\sqrt{K}} \sum_{k=1}^{K} u_k \otimes W_k, \tag{9}$$

which corresponds to a uniform distribution over the randomly chosen $w$.

Now that we have defined the initial state, $\Psi$, for Grover's search we need to define an oracle that marks the vectors inside the version space. Let us define the operator $\hat{\mathcal{F}}_{\phi,y}$ via

$$\hat{\mathcal{F}}_{\phi,y}[u_j \otimes W_0] = (-1)^{1 + f_{w_j}(\phi,y)} [u_j \otimes W_0]. \tag{10}$$

This unitary operation looks at an address vector, $u_j$, computes the corresponding perceptron model $W_j$, flips the sign of any component of the quantum state vector that is in the half space in version space specified by $\phi$ and then uncomputes $W_j$. This process can be realized using a quantum subroutine that computes $f_w$, an application of $V$ and $V^\dagger$ and also the application of a conditional phase gate (which is a fundamental quantum operation that is usually denoted $Z$) [23].

The oracle $\hat{\mathcal{F}}_{\phi,y}$ does not allow us to directly use Grover's search to rotate a quantum state vector that is outside the version space towards the version space boundary because it effectively only checks one of the half–space inequalities that define the version space. It can, however, be used to build an operation, $\hat{G}$, that reflects about the version space:

$$\hat{G}[u_j \otimes W_0] = (-1)^{1 + (f_{w_j}(\phi_1, y_1) \vee \cdots \vee f_{w_j}(\phi_N, y_N))} [u_j \otimes W_0]. \tag{11}$$

The operation $\hat{G}$ can be implemented using $2N$ applications of $\hat{\mathcal{F}}_\phi$ as well as a sequence of $O(N)$ elementary quantum gates, hence we cost a query to $\hat{G}$ as $O(N)$ queries to $\hat{\mathcal{F}}_{\phi,y}$.

We use these components in our version space training algorithm to, in effect, amplify the margin between the two classes from $\gamma$ to $\sqrt{\gamma}$. We give the asymptotic scaling of this algorithm in the following theorem.

**Theorem 3.** *Given a training set that consists of unit vectors $\Phi_1, \ldots, \Phi_N$ that are separated by a margin of $\gamma$ in feature space, the number of queries to $\hat{\mathcal{F}}_{\phi,y}$ needed to infer a perceptron model with probability at least $1 - \epsilon$, $w$, such that $w$ is in the version space using a quantum computer is $N_{\text{quant}}$ where*

$$N_{\text{quant}} \in O\left( \frac{N}{\sqrt{\gamma}} \log^{3/2} \left[ \frac{1}{\epsilon} \right] \right).$$

*Proof.* The proof of the theorem follows directly from bounds on $K$ and the validity of our version space training algorithm. It is clear from previous discussions that the algorithm carries out Grover's search, but instead of searching for a $\phi$ that is misclassified it instead searches for a $w$ in version space. Its validity therefore follows by following the exact same steps followed in the proof of Lemma 2 but with $N = K$. However, since the algorithm need is not repeated $1/\gamma^2$ times in this context we can replace $\gamma$ with 1 in the proof. Thus if we wish to have a probability of failure of at most $\epsilon'$ then the number of queries made to $\hat{G}$ is in $O(\sqrt{K} \log(1/\epsilon'))$. This also guarantees that if any of the $K$ vectors are in the version space then the probability of failing to find that vector is at most $\epsilon'$.

Next since one query to $\hat{G}$ is costed at $N$ queries to $\hat{\mathcal{F}}_{\phi,y}$ the query complexity (in units of queries to $\hat{\mathcal{F}}_{\phi,y}$) becomes $O(N\sqrt{K} \log(1/\epsilon'))$. The only thing that then remains is to bound the value of $K$ needed.

The probability of finding a vector in the version space is $\Theta(\gamma)$ from Theorem 2. This means that there exists $\alpha > 0$ such that the probability of failing to find a vector in the version space $K$ times is at most $(1 - \alpha\gamma)^K \leq e^{-\alpha\gamma K}$. Thus this probability is at most $\delta$ for $K \in \Omega\left(\frac{1}{\gamma} \log(1/\delta)\right)$. It then suffices to pick $K \in \Theta(\log(1/\delta)/\gamma)$ for the algorithm.

The union bound implies that the probability that either none of the vectors lie in the version space or that Grover's search failing to find such an element is at most $\epsilon' + \delta \leq \epsilon$. Thus it suffices to pick

$\epsilon' \in \Theta(\epsilon)$ and $\delta \in \Theta(\epsilon)$ to ensure that the total probability is at most $\epsilon$. Therefore the total number of queries made to $\hat{\mathcal{F}}_{\phi,y}$ is in $O(N \log^{3/2}(1/\epsilon)/\sqrt{\gamma})$ as claimed. $\qquad\square$

The classical algorithm discussed previously has complexity $O(N \log(1/\epsilon)/\gamma)$, which follows from the fact from Theorem 2 that $K \in \Theta(\log(1/\epsilon)/\gamma)$ suffices to make the probability of not drawing an element of the version space at most $\epsilon$. This demonstrates a quantum advantage if $\frac{1}{\gamma} \gg \log(1/\epsilon)$, and illustrates that quantum computing can be used to boost the effective margins of the training data. Quantum models of perceptrons therefore not only provide advantages in terms of the number of vectors that need to be queried in the training process, they also can make the perceptron much more perceptive by making training less sensitive to small margins.

These performance improvements can also be viewed as mistake bounds for the version space perceptron. The inner loop in the version space algorithm attempts to sample from the version space and then once it draws a sample it tests it against the training vectors to see if it errs on any example. Since the inner loop is repeated $O(\sqrt{K} \log(1/\epsilon))$ times, the maximum number of misclassified vectors that arises from this training process is from Theorem 2 $O(\frac{1}{\sqrt{\gamma}} \log^{3/2}(1/\epsilon))$ which, for constant $\epsilon$, constitutes a quartic improvement over the standard mistake bound of $1/\gamma^2$ [16].

## 5   Conclusion

We have provided two distinct ways to look at quantum perceptron training that each afford different speedups relative to the other. The first provides a quadratic speedup with respect to the size of the training data. We further show that this algorithm is asymptotically optimal in that if a super–quadratic speedup were possible then it would violate known lower bounds for quantum searching. The second provides a quadratic reduction in the scaling of the training time (as measured by the number of interactions with the training data) with the margin between the two classes. This latter result is especially interesting because it constitutes a quartic speedup relative to the typical perceptron training bounds that are usually seen in the literature.

Perhaps the most significant feature of our work is that it demonstrates that quantum computing can provide provable speedups for perceptron training, which is a foundational machine learning method. While our work gives two possible ways of viewing the perceptron model through the lens of quantum computing, other quantum variants of the perceptron model may exist. Seeking new models for perceptron learning that deviate from these classical approaches may not only provide a more complete understanding of what form learning takes within quantum systems, but also may lead to richer classes of quantum models that have no classical analogue and are not efficiently simulatable on classical hardware. Such models may not only revolutionize quantum learning but also lead to a deeper understanding of the challenges and opportunities that the laws of physics place on our ability to learn.

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
