[Supplementary Material]

# Supplementary Material for Quantum Perceptron Models

Nathan Wiebe, Ashish Kapoor and Krysta M Svore

*Microsoft Research, Redmond WA 98052*

## I.  PROOFS

Here we provide proofs of several of our results stated in the main body. In particular, we give the proofs of Theorem 2, Lemma 1 and Lemma 2 here.

*Proof of Theorem 2.* Given that the margin of the training set is $\gamma$ there exist a hyperplane $u$ such that $y_i \cdot u^T \Phi_i > \gamma$ for all $i$. If $w$ be a sample from $\mathcal{N}(0, \mathbb{1})$, then lets first compute what is the probability that perturbing the maximum margin classifier $u$ by amount $w$ would lead still lead to a perfect separation. If we consider a data point $\Phi^*$ that lies on the margin, i.e. $y_i \cdot u^T \Phi^* = \gamma$, we are interested in the probability that $y_i \cdot (u+w)^T \Phi^* > 0$ and $y_i \cdot (u+w)^T \Phi^* < 2\gamma$. The first inequality corresponds to preventing misclassification of $\Phi^*$, while the second one corresponds to preventing misclassification of the point belonging to the other class and on the margin. This is same as asking what is the probability that:

$$-\gamma < y_i \cdot w^T \Phi^* < \gamma \tag{1}$$

Let us define $z_i := y_i \cdot w^T \Phi^*$ . Since $w \sim \mathcal{N}(0, \mathbb{1})$ and $\|\Phi\| = 1$ we can show that $z_i \sim \mathcal{N}(0,1)$. Thus, we can write the probability that $-\gamma < z_i < \gamma$ as:

$$P(-\gamma < z_i < \gamma) = \mathrm{erf}\left(\frac{\gamma}{\sqrt{2}}\right) \tag{2}$$

Here $\mathrm{erf}(z) = \frac{1}{\sqrt{\pi}} \int_{-z}^{z} e^{-\frac{x^2}{2}} dx$ is the error function for the standard normal distribution. Since $\Phi^*$ is on the margin, the probability that the sample $w$ will lie in the version space can be simply characterized as the above probability $P(-\gamma < z_i < \gamma)$. It is straightforward to show using Maclaurin series expansion that:

$$P(w \in \mathrm{VS}) = \frac{2}{\sqrt{\pi}} \left( \frac{\gamma}{\sqrt{2}} - \frac{\gamma^3}{2^{3/2}3} + \frac{\gamma^5}{2^{5/2}10} - \frac{\gamma^7}{2^{7/2}42} \cdots \right) \tag{3}$$

Note, that in our case $\Phi_i$ are unit normalized for all $i$, thus $\gamma < 1$. Which in turn implies that most of the higher order terms will be close to zero in the limit of small $\gamma$ and:

$$P(w \in \mathrm{VS}) = \frac{\gamma}{\sqrt{2\pi}} + O(\gamma^3), \tag{4}$$

which proves our theorem for $\gamma < 1$. $\qquad\square$

*Proof of Lemma 1.* There exists a simple algorithm for achieving this upper bound. Draw $N\lceil\log(1/\epsilon\gamma^2)\rceil$ samples from the set of training vectors. If any are misclassified perform the update, otherwise report that the model classifies all the data.

The proof of validity of this algorithm is trivial and the success probability claim is also quite simple. Given that we draw $k$ samples from the distribution the probability that any of them fail to detect a mistake, given such a mistake exists, is at most

$$(1 - 1/N)^k \le \exp(-k/N). \tag{5}$$

If we want this error to be at most $\delta$ then it suffices to take

$$k = \lceil N \log(1/\delta) \rceil. \tag{6}$$

One query to $f_w$ is required per $k$, which means that $k$ is also equal to the query complexity. Thus if at least one mistake occurs then the algorithm will find it with the aforementioned probability if $\delta = \epsilon\gamma^2$. If such an example does not exist, then the algorithm will correctly conclude that a separating hyperplane has already been found. Therefore in either case the success probability is at least $1 - \epsilon\gamma^2$ as required. $\qquad\square$

---

**Algorithm 1** Online quantum perceptron training algorithm

---

**for** $k = 1, \ldots, \lceil \log_{3/4} \gamma^2 \epsilon \rceil$ **do**
**for** $j = 1 : \lceil \log_c(1/\sin(2\sin^{-1}(1/\sqrt{N}))) \rceil$ **do**
Draw $m$ uniformly from $\{0, \ldots, \lceil c^j \rceil\}$.
Prepare quantum state $\Psi$.
$\Psi \leftarrow ((2\Psi\Psi^\dagger - \mathbb{1})F_w)^m \Psi$.
Measure $\Psi$, assume outcome is $u_q$.
$(\phi, y) \leftarrow U^c(q)$.
**if** $f_w(\phi, y) = 1$ **then**
Return $w' \leftarrow w + y\phi$
**end if**
**end for**
**end for**
Return $w'$

---

*Proof of Lemma 2.* In order to see this, let us first examine the inner loop of Algorithm 2, which involves performing the update $\Psi \leftarrow ((2\Psi\Psi^\dagger - \mathbb{1})F_w)^m \Psi$. We know from our discussion of Grover's algorithm in the main body that if we define the initial probability of successfully find a mistake to be $\sin^2(\theta_a)$ then the probability of finding a $j$ such that $\mathcal{F}_w \Phi_j = -1$ after $m$ updates is $\sin^2((2m+1)\theta_a)$. Since this corresponds to finding a vector that the perceptron fails to classify properly, these steps amplify the probability of finding a perceptron error. The query to $U^c$ that follows identifying the index of this training vector then converts this result into a classical bitstring that can then be used to perform a perceptron update. Therefore the inner loop performs a perceptron update with probability $\sin^2((2m+1)\theta_a)$ using $m$ queries to $F_w$.

Under the assumption that $c \in (1, 2)$ the next loop repeats this sampling process until $m \geq M_0$ in order to ensure that the probability of finding a misclassified element is at least $1/4$ [1]. This can be seen using the following argument. First we need to show that the exponential search heuristic requires $O(M_0)$ queries. Each iteration of the middle loop requires requires a number of queries that is at most proportional to $\lceil c^i \rceil$. Therefore the total number of queries is at most proportional to

$$
\begin{aligned}
\sum_{i=0}^{\lfloor \log_c M_0 \rfloor} \lceil c^i \rceil &\leq \frac{c^{\lfloor \log_c M_0 \rfloor + 1}}{c-1} + \lceil \log_c M_0 \rceil \\
&\leq \frac{c}{c-1} M_0 + \lceil \log_c M_0 \rceil.
\end{aligned}
\tag{7}
$$

Given $c$ is a constant we have that $(c-1) \in \Theta(1)$ and thus $\sum_{i=0}^{\lfloor \log_c M_0 \rfloor} c^i \in O(1/\sin(\theta_a))$ from (4) in the main body. If there exists an element that the algorithm makes a mistake on then $\theta_a \geq \sin^{-1}(\sqrt{1/N}) \in \Omega(1/\sqrt{N})$ because the lowest probability of success corresponds to the case where there is only one training vector that is misclassified out of $N$. From this we see that, if a misclassified vector exists, then the middle loop is repeated at least $\log_c(M_0)$ times which means that the final iteration taken corresponds to $m \geq M_0$ for the purposes of (6) in the main body. Therefore, under these assumptions, the probability that the middle loop updates the perceptron weights is at least $1/4$ from [1]. Given that a mistake exists to be found the middle loop outputs such an element with a probability of failure that is at most $3/4$ from (6) in the main body. Furthermore, $O(M_0) = O(1/\sqrt{N})$ queries to $F_w$ are required by the inner loop.

The outer loop serves to amplify the success probability to at least $1 - \epsilon$ from the (average) success probability for $m \geq M_0$, which is at least $1/4$, given that the perceptron makes a mistake on at least one training vector [1]. Let us assume that we repeat the middle loop of Algorithm 1 $k$ times past this point and terminate searching for a marked state if the probability of failing to detect the element is at most $\delta$. Since the probability of the middle loop failing to find such an element, given that it exists, is at most $3/4$ the probability of failing to find a marked state all $k$ times is at most $(3/4)^k$ which implies that it suffices to choose

$$
k = \lceil \log_{3/4}(\delta) \rceil.
\tag{8}
$$

Given this error bound, the number of Grover iterations needed for the algorithm to find the marked element is

$$
O(\sqrt{N} \log_{3/4} \delta) \in O(\sqrt{N} \log(1/\delta)).
$$

The result then follows by taking $\delta = \epsilon\gamma^2$. Therefore the lemma holds if $\theta_a > 0$.

If $\theta_a = 0$ then the algorithm will never find a quantum state vector that the perceptron misclassifies and will successfully conclude that there is not a marked state after $O(\sqrt{N} \log(1/\epsilon\gamma^2))$ queries. Therefore the lemma also holds in the trivial case. $\qquad\square$

---

**Algorithm 2** Quantum version space perceptron training algorithm

---

**for** $k = 1, \ldots, \lceil \log_{3/4} \epsilon \rceil$ **do**

**for** $j = 1 : \lceil \log_c(1/\sin(2\sin^{-1}(1/\sqrt{K}))) \rceil$ **do**

Draw $m$ uniformly from $\{0, \ldots, \lceil c^j \rceil\}$.

Prepare quantum state $\Psi = \frac{1}{\sqrt{K}} \sum_{p=1}^{K} u_p \otimes W_0$.

$\Psi \leftarrow ((2\Psi\Psi^\dagger - \mathbb{1})G)^m \Psi$.

Measure $\Psi$, assume outcome is $u_q$.

$w \leftarrow V^c(q)$.

**if** $f_w(\phi_\ell, y_\ell) = 0$ for all $\ell \in \{1, \ldots, N\}$ **then**

Return $w$

**end if**

**end for**

**end for**

Return $w = 0$

---

## II. ALGORITHMS

Here we provide explicit algorithms for our feature space and version space perceptron training algorithms. The purpose of Algorithm 1 is to give a formal statement of the online version of the perceptron training algorithm, which gives a reduced cost of training if the quantum data is provided in a superposition of states. The second algorithm gives our version space analogue, which is not an online training algorithm but nonetheless is significant because we can prove bounds on the number of mistakes made in the training process.

## Footnotes

[1] Gilles Brassard, Peter Hoyer, Michele Mosca, and Alain Tapp. Quantum amplitude amplification and estimation. *Contemporary Mathematics*, 305:53–74, 2002.