[Reviews · NeurIPS 2016]

Reviewer 1

Summary

The authors argue that quantum computing can (a) speed up perceptron training significantly and (b) can improve on performance bounds.

Qualitative Assessment

I find this paper difficult to comprehend. This also applies to the motivation and conclusions drawn from this work. It is argued, for instance, that quantum computing can speed up standard perceptron training significantly. I cannot follow the authors in that that this would render the perceptron a "quantum perceptron" or even may "lead to a deeper understanding of ... our ability to learn".

Confidence in this Review

1-Less confident (might not have understood significant parts)


Reviewer 2

Summary

The paper rigorously studies a question: what kind of advantages can we expect from quantum computing in machine learning? This is a difficult question to ask, as in many learning algorithms we either have a clear idea of their computational complexity or their statistical complexity, but seldom both. The authors focus on the well-understood perceptron to study the limits of quantum computers in machine learning. The scheme uses and integer discretization the usual floating-point representation and assumes a quantum random access memory. The ideas pivot on a Grover-type search.

Qualitative Assessment

The paper is meticulously written and it reads well. Little prior knowledge in the basics of quantum mechanics is sufficient to follow the argument. The classical weight update scheme is clearly simplistic, but the authors provide ample references for other algorithms, and this simple algorithm allows for a clean and straightforward comparison with the quantum case. Grover's search in machine learning is certainly not a ground-breaking idea, but investigating the bounds in a rigorous way has been lacking. The only flaw I can point out is the lack of references to other works on quantum perceptrons. Most notably, Lewenstein (1994) investigated quantum perceptrons from the perspective of statistical learning theory, which is the closest in spirit to the present work. M. Lewenstein (1994). Quantum Perceptrons, Journal of Modern Optics, 41:12, 2491-2501, DOI: 10.1080/09500349414552331.

Confidence in this Review

3-Expert (read the paper in detail, know the area, quite certain of my opinion)


Reviewer 3

Summary

The authors present two ways of training the classical perceptron algorithm on a quantum computer based on Grover's search algorithm. The key idea behind their algorithms is the mapping of the training procedure to a search for misclassified datapoints or separating hyperplanes (in the version space) from a uniform distribution over the data and a defined number of sampled hyperplanes, respectively. This mapping is explained in detail. The authors formally show that their algorithms yield a quadratic speedup in the number of necessary training data points and the margin between the two classes, respectively.

Qualitative Assessment

In my opinion, this paper deserves the attention of the NIPS audience at least on poster level for sound investigation of an highly interdisciplinary and fundamental question whose answer might eventually have a strong impact on ML. To the best of my knowledge, the proofs and arguments presented are correct. While an experimental evaluation would of course have been very good to back up the theoretical claims, the absence of a device capable of executing the algorithm excuses the authors from my point of view. As far as quantum algorithms go, the results are hence well presented and analyzed. It is clear that this paper stands out in terms of originality and novelty (at least for the NIPS audience). While Grover's search is not a new algorithm, this application of it and the corresponding interdisciplinary insight certainly are novel by NIPS standards. The impact is perhaps the most critical point as, for the reason stated above, this work is of purely theoretical nature and also far away from what most machine learning researches probably work on. However, it certainly does bridge multiple disciplines and should a quantum computer be built in the future, this work will surely have a strong impact and spawn much future work. Even if the immediate impact might not be high, researchers should be interested in important future developments in their field for which this work poses an example. From my perspective, the authors (apart from some superfluous words that should be deleted and referring to Algorithm 2 in the proof of Lemma 2 when it should be Algorithm 1) try to express their ideas in a language understandable for the NIPS audience and rely on as little previous knowledge about quantum mechanics as possible. The explanations and proofs are also almost exclusively based on purely mathematical arguments, so that they should be understandable. However, some concepts such as 'superposition' and resulting speedups might have deserved more explanation. The lack of knowledge of basic principle in QM might hinder the understanding of some readers. However, given the page limit I find that the authors did reasonably well in introducing the most important concepts from quantum mechanics.

Confidence in this Review

2-Confident (read it all; understood it all reasonably well)


Reviewer 4

Summary

The paper applies quantum computation to perceptron learning in fundamental ways and obtained improved complexity (from O(N) to O(sqrt(N))) and error bounds from (O(1/gamma^2) to O(1/gamma)). Both of these improvements are inspired by Grover's search/amplitude amplification, which informally searches through N unsorted elements in only O(sqrt(N)) time.

Qualitative Assessment

Applying quantum methods to machine learning is a very interesting problem, and the results obtained in this paper is nice. Just like Grover's search differs funcamentally from classical search algorithms, the authors try to use quantum computation in a fundamental way, instead of replacing one subroutine with another quantum one while staying within the classical framework.

Confidence in this Review

1-Less confident (might not have understood significant parts)


Reviewer 5

Summary

The authors propose a perceptron model learning strategy using quantum computing that provides significant training speedups. Besides the actual contribution on the line of the perceptron model, the authors show how re-thinking learning strategies from a quantum computing point of view may obtain better results rather to trying to transfer the traditional learning procedure to a quantum paradigm.

Qualitative Assessment

Major corrections: * My major concern is related to the assumption that training observations have unit norm. The main contribution of this work relies on the training speedup by using quantum computing. However, Theorems 1, 2 and 3 and Lemma 2 assume that input vectors have unit length. This is a major assumption from a machine learning point of view and will only hold for per-observation normalization schemes. For instance, in a one-dimensional setting, this approach only allows two observations (-1 and 1). Does this assumption affect your conclusions? Would you obtain the same results in other scenarios?. * The encoding explanation presented in lines 133-134 should be improved. I didn't understand the final representation [0,0,0,0,0,1]. * Some phrases are unclear (e.g. lines 115-116, while-but). Please verify grammar and spelling. Minor corrections: Line 83: algorthm -> algorithm. Line 95: quatum -> quantum

Confidence in this Review

1-Less confident (might not have understood significant parts)


Reviewer 6

Summary

Developed two quantum algorithm for perception learning.

Qualitative Assessment

1 Significant speedup. 2 Improved classical mistake bound.

Confidence in this Review

1-Less confident (might not have understood significant parts)